# Understanding the influence of ethnicity on adherence to antidiabetic medication: Meta-ethnography and systematic review

Rayah Asiri[1,2☯], Anna Robinson-Barella[1☯], Anum Iqbal[1☯], Adam Todd[1☯], Andy Husband[1☯]*

1 School of Pharmacy, Newcastle University, Newcastle upon Tyne, United Kingdom, 2 School of Pharmacy, King Khalid University, Abha, Saudi Arabia

☯ These authors contributed equally to this work.
* andy.husband@newcastle.ac.uk

## Abstract

### Introduction

A high prevalence of diabetes and diabetes-related complications in people from minority ethnic communities in high income countries is of significant concern. Several studies have indicated low adherence rates to antidiabetic medication in ethnic minority groups. Poor adherence to antidiabetic medication leads to a higher risk of complications and potential mortality. This review aims to qualitatively explore the barriers to and facilitators of adherence to antidiabetic medication among ethnic minority groups in high-income countries.

### Methods

A comprehensive search of MEDLINE, Embase, CINAHL, PsycINFO, and Global Health databases for qualitative studies exploring the barriers to or facilitators of adherence to antidiabetic medication in minority ethnic groups was conducted from database inception to March 2023 (PROSPERO CRD42022320681). A quality assessment of the included studies was conducted using the Critical Appraisal Skills Programme (CASP) tool. Key concepts and themes from relevant studies were synthesised using a meta-ethnographic approach. The Grading of Recommendations Assessment, Development and Evaluation Confidence in the Evidence from Reviews of Qualitative research (GRADE-CERQual) approach was used to assess the Confidence in the review findings.

### Result

Of 13,994 citations screened, 21 studies that included primary qualitative studies were selected, each of which involved people from minority ethnic communities from eight high income countries. This qualitative evidence synthesis has identified three overarching themes around the barriers to and facilitators of adherence to antidiabetic medication among ethnic minority groups.: 1) cultural underpinnings, 2) communication and building relationships, and 3) managing diabetes during visiting home countries. Based on the GRADE-CERQual assessment, we had mainly moderate- and high-confidence findings.

**Data Availability Statement:** All relevant data are within the paper and its Supporting Information files.

**Funding:** This work was supported by the Saudi Arabian Cultural Bureau in the United Kingdom and King Khalid University in Saudi Arabia. There was no additional external funding received for this study. The funders had no role in study design, data collection and analysis, decision to publish, or preparation of the manuscript.

**Competing interests:** All authors declare that they have no conflicts of interest.

## Conclusion

Multiple barriers and facilitators of adherence to antidiabetic medication among people from minority ethnic communities in high-income countries have been identified. A medication adherence intervention focusing on identified barriers to adherence to antidiabetic medication in these communities may help in improving diabetes outcomes in these groups.

## Introduction

Diabetes is a significant global public health issue, resulting in serious and costly complications and reduced life expectancy [1, 2]. There are several types of diabetes, including type 1, type 2, maturity-onset diabetes of the young, gestational diabetes, neonatal diabetes, and secondary causes due to endocrinopathies and steroids use [3]. It is a chronic disease affecting approximately 536 million people worldwide in 2021, and this number is expected to increase to 783.2 million by 2045 [1]. The prevalence of diabetes and related complications are higher among people from minority ethnic communities. In the United Kingdom, type 2 diabetes is around three to five times more prevalent amongst minority groups, particularly Asian and African-Caribbean groups, when compared to White populations [4]. Similarly, in the United States, Native Americans, non-Hispanic Blacks, and Hispanic communities report a higher prevalence of diabetes compared to non-Hispanic White people [5, 6]. Additionally, minority ethnic groups in the United States are more likely to suffer diabetes-related complications, including microvascular problems, compared to White groups [6, 7]. The risk of developing macrovascular complications is higher in people from ethnic minority communities in the UK, particularly among Asian groups [8, 9].

Medication adherence affects the effectiveness of the treatment and management of diabetes. Associations in adherence and improved long-term health outcomes for people living with diabetes have been well-reported [10–12]. Yet, despite the importance of medication adherence in achieving optimal clinical outcomes, evidence remains that certain patient groups continue to report poor adherence to antidiabetic medications; in particular, those from minority ethnic communities. In a number of high-income countries such as the United States, the United Kingdom, New Zealand, Singapore, and Canada, lower rates of adherence to antidiabetic medications have been reported amongst minority ethnic groups [13–17]. To optimise diabetes care among people from ethnic minority communities, it is crucial to gain a better understanding of patient-related barriers and facilitators associated with adherence to antidiabetic medications. A number of qualitative studies have examined the experience of people from minority ethnic communities regarding diabetes self-management generally, or their adherence to antidiabetic medication specifically [18–20]. Although individual qualitative studies can be valuable for providing insights about factors influencing adherence to antidiabetic medications in minority populations, qualitative evidence synthesis can lead to a greater depth of understanding beyond individual qualitative research findings, which is where this study seeks to contribute [21].

To date, the published qualitative evidence reviews have examined adherence to diabetes self-management among people with type 2 diabetes [22], adherence to particular antidiabetic medications [23, 24], and adherence to diabetes self-management among people from one ethnic group with type 2 diabetes [25]. This review utilized a meta-ethnographic qualitative synthesis approach to explore the barriers to, and facilitators of, adherence to antidiabetic medications experienced by people from minority ethnic communities in high-income countries. The meta-ethnographic approach enables reviewers to re-interpret conceptual data from

primary qualitative studies (*i.e.*, original themes) to enable the production of a new conceptual evidence synthesis using the findings of individual studies [26, 27]. As a result, future policy and guidance might be informed and shaped by this re-interpretation [27].

## Methods

This meta-ethnographic systematic review has been conducted according to the PRISMA (Preferred Reporting Items for Systematic Reviews and Meta-Analyses) guidelines(S1, S2 Tables in S1 File) [28] and is registered with PROSPERO (registration number CRD42022320681).

### Search strategy and information sources

Five electronic databases–MEDLINE, Embase, CINAHL, PsycINFO, and Global Health–were searched systematically from their inception to March 2023. No limit was applied on the language or the date of publication. To identify all relevant publications, grey literature(*via* searching OpenGrey, the top 150 Google search hits, and EThOS), and hand searches of the reference lists of all included studies and relevant systematic reviews, and the citations of all included studies, using the citations provided by Google scholar were performed. The databases were searched using a combination of diabetes, medication adherence, ethnicity, and qualitative research search terms. The search strategy is described in (S1 Text).

### Eligibility criteria

The inclusion criteria for this review specified studies that explored factors perceived as barriers to and/or facilitators of adherence to antidiabetic medication among minority ethnic communities (defined as populations who are numerically smaller and have a different ethnic, religious, or linguistic background than the majority population in the country where the study was conducted) [29], with type 1 or type 2 diabetes, conducted in high-income countries. This was determined according to the 2022 Gross National Income (GNI) per capita, with high-income countries being those with a GNI per capita of more than USD 13,205 [30]. The studies must also include qualitative data about the views of ethnic minority groups and they must have been conducted using a qualitative methodology. In the case of studies conducted in "mixed" countries (high-income and low or middle-income countries), the data exclusively from the high-income countries were included.

Exclusion criteria included studies that explored factors perceived as barriers to and/or facilitators of adherence to diabetes self-management that did not focus on medication adherence (such as physical activity, diet, self-monitoring of blood sugar); those studies that were conducted in low- and middle-income countries (low-income countries are those with a GNI per capita of less than USD 1,085, and middle-income countries are those with a GNI per capita of greater than USD 1,086, but less than USD 13,205) [30]; studies with qualitative data about the views of majority and minority groups without labelling data by ethnic minority groups; and study types that were mixed-method and quantitative studies as the meta-ethnographic approach's exclusively focus on inclusion of qualitative studies [31], systematic reviews, conference abstracts, and clinical trials were also excluded.

### Study selection and screening

All citations were exported to EndNote 20 reference manager software to manage duplicate studies, and the screening process [32]. One reviewer (RA) screened the titles and abstracts from the database searches in accordance with the eligibility criteria. The full texts of articles that met the inclusion criteria and those that could not be definitively rejected were retrieved. The screenings

of the articles' full text were undertaken by RA and checked independently in full by authors AKH or AR-B. Any disagreement was resolved through discussion and consensus.

## Study reading, data extraction and quality appraisal

The included studies were read by two reviewers (RA and AR-B) to ensure familiarity with the data. The data extraction was performed by RA using a customised data extraction form (S7 Table in S1 File) (designed by RA). Data extracted included information about the author details, study aim, setting, participant demographics and study findings, including original participant quotes and author interpretations. Quality assessment was conducted by (RA and AI) using the Critical Appraisal Skills Programme (CASP) tool for qualitative research [33].

## Analysis and interpretive synthesis

This review used the meta-ethnographic approach described originally by Noblit and Hare [34], and commonly used in healthcare research [26, 27, 35]. The seven phases of the meta-ethnography approach [34] are listed in (Fig 1).A meta-ethnography is an inductive and interpretive approach that allows researchers to analyse, transfer, and understand ideas, themes, and metaphors across a number of studies to better understand or inform broader concepts [34, 36]. This review aimed to understand how ethnicity influences adherence to antidiabetic medication by exploring the barriers and facilitators of adherence to medication among people from minority ethnic communities. Meta-ethnography was chosen to understand how ethnicity affects adherence to antidiabetic medications and build a conceptual model that explains the barriers and facilitators of adherence in ethnic minority communities. Meta-ethnography encourages a conceptional approach to understanding experience by synthesising primary research findings rather than describing the results of previous studies [34].

**Determining how studies are related.** To determine how studies are related, a table of findings from each primary study was prepared (including the concepts and metaphors developed by the original authors); this facilitated exploration and comparison of the findings. Two reviewers (RA and AR-B) discussed the relationships between metaphors across studies.

**Translating the studies.** Due to similar and reoccurring themes across the studies, a reciprocal translational was used to develop third-order constructs. Reciprocal translation involves a systematic comparison where the findings from the first study are compared with the second study, and shared overarching themes are combined [26, 34]. The combined themes from these two studies are then compared and combined with the recurring themes in the third study, and so on. By doing so, the authors identified, interpreted, and consolidated common themes that emerged from the studies. Whenever disagreements arose about the identification or interpretation of a theme, the reviewers re-read the studies and discussed their points of view until they reached an agreement.

**Synthesising the translations and expression the synthesis.** After completing the translation of the studies into one another, the authors (RA and AR-B) began synthesising the data by carefully analysing the first-order constructs (*i.e.*, participant quotations) and the second-order constructs (*i.e.*, authors interpretations) to arrive at the third-order constructs (overall interpretations). Finally, these third-order constructs represented the findings of the current meta-ethnography. Four overarching themes (termed third-order constructs) and subsequent subthemes were developed; these are consistent with, but extend beyond, the original study results.

## Confidence in the synthesised findings

The confidence in this review's findings was assessed by the GRADE-CERQual approach (Confidence in the Evidence from Reviews of Qualitative Research) [37]. A GRADE-CERQual

## Phase 1: Getting started

## Phase 2: Deciding what is relevant to the initial interest

## Phase 3: Reading the studies

## Phase 4: Determining how studies are related

## Phase 5: Translating the studies into one another

## Phase 6: Synthesizing translations

## Phase 7: Expressing the synthesis.

**Fig 1. The seven phases of meta-ethnography [31].**

assessment considers four main components: the methodological limitations of included studies, the coherence of the qualitative evidence synthesis findings, the adequacy of data used to support the review finding, and the relevance of the included studies to the review question. Based on the results of CERQual, we classified overall confidence into three levels: high, moderate, and low, and presented the results in a summary of findings table.

## Results

### Search results

A total of 13,994 citations were identified through database searches and other sources, including the grey literature and bibliographies of the included studies. Following the removal of duplicate publications, 12,987 papers were screened for eligibility. After the title and abstract screening, a total of 69 records met the eligibility criteria for full-text checking. During the full-text review, 48 studies were excluded for reasons outlined in the PRISMA flowchart in

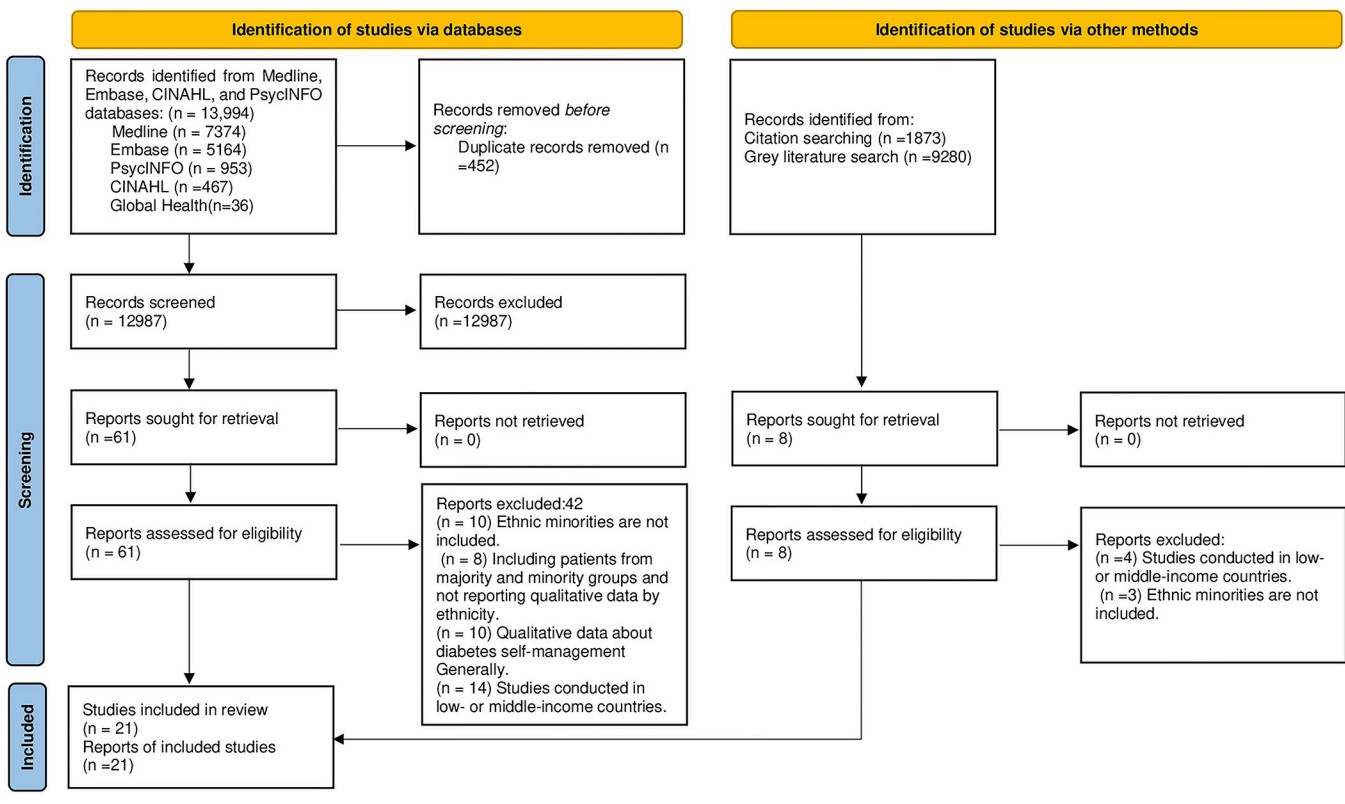

**Fig 2. PRISMA flow diagram of included studies.**

(Fig 2). In total, 21 studies were eligible for inclusion for this meta-ethnographic systematic review.

## Study characteristics

All studies were published between 2005 and 2022 and were conducted in eight countries: the United Kingdom (n = 6) [18, 38–42], the United States (n = 8) [20, 43–49], Australia (n = 2) [50, 51], New Zealand (n = 1) [52], Canada (n = 1) [53], Belgium (n = 1) [54], Qatar (n = 1) [19], and multicenter study (Amsterdam, Berlin and London) [55]. The sample sizes ranged from 5 [53] to 84 participants [46], and the mean age of patients ranged from 49.1(SD: 9.9) [45] to 68.5(SD: 2.5) [44] years. Minority ethnic groups that were reported across the included studies were: Asian, African American, Mexican American, African Caribbean, Turkish, Māori,Black African,Latino, Slavic and Pacific. Twelve studies focused on exploring the barriers to, and/or facilitators of, adherence to antidiabetic medication [19, 20, 42, 43, 45, 48–54]; the remaining studies examined the barriers to and/or facilitators of diabetes self-management more broadly, including medication adherence [18, 38–41, 44, 46, 47, 55]. Study characteristics are detailed in (Table 1).

## Study quality

According to the Critical Appraisal Skills Programme (CASP) checklist, 17 studies met the criteria related to the clarity of the research aims, the ethical issue consideration, recruitment, data collection, analysis, and reporting of study results. However, 19 of the studies did not provide sufficient detail about the relationship between the researcher and the participants. The

**Table 1. Characteristics of the included studies.**

| Author & Year | Study aim (extracted verbatim from the original study) | Setting | Participants | | | | | | | Method of recruitment, data collection & analysis |
|---|---|---|---|---|---|---|---|---|---|---|
| | | | Sample size | Gender/sex reported by participants | Age(mean ±SD)/ median (IQR) | Ethnic minority group: | Type of diabetes | Antidiabetic medication | |
| Jamil *et al.* (2022) [49] | 'To explore how cultural perspectives of South Asian immigrants in the U.S. impact adherence to medications for DM2 and CVD.' | Birmingham Free Clinic and the UPMC Matilda Theiss Health Center, two safety net provider settings in Pittsburgh US | 12 participants. 7 Pakistani 4 Indian 1 Bangali | Female: n = 6(50) Male: n = 6 (50) | 63(49–75) | South Asian | Type 2 diabetes | Not mentioned | Semi-structured interviews Data analysis was not reported clearly |
| Timsina *et al.* (2022) [48] | 'To understand factors affecting diabetes medication adherence in Bhutanese Refugees.' | University of Kansas Medical Center US | 13 participants. | Female: n = 8(62) Male: n = 5 (38) | 58(42–79) | Bhutanese people | Type 2 diabetes | Insulin and/or oral antidiabetic drugs | Focus groups Content analysis. |
| Parkin *et al.* (2021) [52] | 'To explore the views of Māori, Pacific, and non-Māori non-Pacific patients with type 2 diabetes about what helps and hinders metformin adherence and persistence after initiating therapy.' | Primary care providers in Auckland, Wellington, and Dunedin. New Zealand | Total: 30 participants. 10 Māori, 10 Pacific, and 10 non-Māori non-Pacific participants. | Female: n = 22(73) Male: n = 8 (27) | Not reported. | Māori Pacific Non-Māori non-Pacific | Type2 diabetes | Metformin | Purposive sample. Semi-structured, face-to-face interview. Thematic analysis, using the Theory of Planned Behaviour |
| Ahmad *et al.* (2021) [50] | 'To investigate patients' medication-taking behaviour and factors that influence adherence, particularly at its three phases among Indian migrants.' | Public places in Sydney. Australia | 23 participants. | Female: n = 5(22) Male: n = 18 (78) | 39(33–72) | Indian migrants. | Type2 diabetes | Oral antidiabetic medications, insulin, and/or Ayurvedic medicines | Semi-structured face-to-face interviews. Thematically analysed using a framework for thematic analysis. |
| Omodara *et al.* (2021) [38] | 'To examine cultural beliefs, attitudes, and practices of Black sub-Saharan Africans (BsSAs) in the UK regarding their type 2 diabetes (T2D) self-management using the concepts of the PEN-3 cultural model.' | Black sub-Saharan Africans (BsSA) communities. UK | 36 participants. | Female: n = 25(69) Male: n = 11 (31) | 52.2±7.2 | Black sub-Saharan African | Type2 diabetes | Insulin and/or oral antidiabetic drugs | Purposive sampling. Semi-structured interviews and field notes. Narrative thematic analysis using PEN-3 cultural model. |
| Pardhan *et al.* (2020) [18] | 'To determine whether barriers to diabetes awareness and self-help differ in South Asian participants of different demographic characteristics (age, gender, and literacy) with type 2 diabetes living in the United Kingdom.' | Community centers and research facilities in Peterborough and Cambridge. UK | 35 participants. 26 Pakistani 5 Nepalese 4 Indian | Female: n = 17(49) Male: n = 18 (51) | 52±2.9 | South Asian | Type2 diabetes | Not mentioned | Pragmatic sampling Focus group Thematic analysis |

*(Continued)*

**Table 1.** (Continued)

| Author & Year | Study aim (extracted verbatim from the original study) | Setting | Participants | | | | | | | Method of recruitment, data collection & analysis |
|---|---|---|---|---|---|---|---|---|---|---|
| | | | Sample size | Gender/sex reported by participants | Age(mean ±SD)/ median (IQR) | Ethnic minority group: | Type of diabetes | Antidiabetic medication | | |
| de-Graft Aikins *et al.* (2019) [55] | 'To assess perceptions and knowledge of T2D among Ghanaian migrants in Europe, and their compatriots in Ghana (including gaps in perceptions and knowledge that might raise the risk of T2D, and to examine how insights on perceptions and knowledge could be used to develop appropriate T2D interventions for these communities.' | Amsterdam, London and Berlin | 66 participants. 14 Amsterdam 12 London 40 Berlin | Amsterdam: Female: n = 3(21) Male: n = 11 (79) London: Female: n = 8(67) Male: n = 4 (33) Berlin: Female: n = 28(70) Male: n = 12 (30) | Not reported | Ghanaian migrants | Type2 diabetes | Not mentioned | | Focus groups. Thematic analysis. |
| Jaam *et al.* (2018) [19] | 'To explore the barriers to medication adherence among patients with uncontrolled diabetes within primary care by integrating the perspectives of the patients and their health care providers.' | Two primary health care Center Qatar | 14 Participants 7(ethnic minority group) | Female: n = 4(29) Male: n = 10 (71) | 58.3±8.1 | 2 Indian 2 Sri Lankan. 1 Pakistani 1 Iranian | Diabetes (not specified the type) | Insulin and/or oral antidiabetic drugs | | Purposive sample Semi-structured face-to-face interviews. Thematic analysis technique. |
| Shiyanbola *et al.* (2018) [20] | 'To explore the reasons for medication nonadherence and adherence among AAs with type 2 diabetes and understand their perceptions of the solutions for enhancing adherence.' | Community center, church, apartment building, or senior center. Two different cities within the state (Suburbs and Urban City) US | 40 participants. | Female: n = 24(61) Male: n = 16 (39) | 53±4.94 | African American | Type 2 diabetes | Not mentioned. | | Purposive sample Focus group Qualitative content analysis |
| Sapkota *et al.* (2018) [51] | 'To explore medication-taking behaviour in Nepalese patients with type 2 diabetes, specifically investigating anti-diabetic medication initiation and implementation and their reasons for cessation and persistence with therapy.' | Public places Australia and Nepal. | 48 participants. 18 participants in Sydney. 30 participants in Kathmandu | Female: n = 6 (33.3) Male: n = 12 (66.7) | 54.2(24–73) | Nepalese | Type 2 diabetes | Anti-diabetic medication | | Snowball sampling Face-to-face interview. Thematic analysis. |
| Bockwoldt *et al.* (2017) [43] | 'To describe the experiences of taking diabetes medications among midlife (35–59) African American men and women with type 2 diabetes and to identify factors that influence these experiences.' | Hospital-based clinic. Chicago US | 15 participants. | Female: n = 9(60) Male: n = 6 (40) | 51.7±5.7 | African American | Type 2 diabetes | Insulin and/or oral antidiabetic drugs | | Purposive sample Semi-structured interviews. Thematic analysis |

*(Continued)*

**Table 1.** (Continued)

| Author & Year | Study aim (extracted verbatim from the original study) | Setting | Participants | | | | | | Method of recruitment, data collection & analysis |
|---|---|---|---|---|---|---|---|---|---|
| | | | Sample size | Gender/sex reported by participants | Age(mean ±SD)/ median (IQR) | Ethnic minority group: | Type of diabetes | Antidiabetic medication | |
| Patel *et al.* (2016) [39] | 'To explore the role of social networks and beliefs about diabetes in British South Asians, to better understand their management behaviours whilst holidaying in the East.' | Greater Manchester UK | 44 participants. | Female: n = 21(48) Male: n = 23 (52) | 61±12.5 | British South Asians: Indian Pakistani Bangladeshi Nepalese | Type 1 or Type 2 diabetes | Insulin and/or oral antidiabetic drugs | Random sampling & Purposive sampling. Semi-structured face-to-face interviews. Thematic analysis using constant comparison approach |
| Joo & Lee, (2016) [44] | 'To explore one research question: What barriers to and facilitators of type 2 diabetes (T2DM) self-management exist for elderly Korean American immigrants who live in the Midwest.' | US | 23 participants. | Female: n = 11(48) Male: n = 12 (52) | 68.5±2.5 | Korean American immigrant | Type 2 diabetes | Insulin and/or oral antidiabetic drugs | Convenience and purposive sampling. Focus group. Standard content-based analysis. |
| Peeters *et al.* (2015) [54] | 'To explore perspectives of Turkish migrants with type 2 diabetes mellitus on adherence to oral hypoglycaemic agents.' | Primary care and community sources in Ghent, Belgium | 21 participants. | Female: n = 12(57) Male: n = 9 (43) | Not reported. | Turkish | Type 2 diabetes | Insulin and/or oral antidiabetic drugs | Theoretical sampling procedure. In-depth interview Grounded theory approach |
| Mohan *et al.* (2013) [45] | 'To conduct a qualitative evaluation to: 1) better understand medication taking practices among low-income Latinos; and 2) evaluate Latino patients' perceptions regarding PictureRx, specifically assessing whether PictureRx could address patients' challenges with medication taking and be a culturally appropriate tool to improve communication about medication information.' | Two safety net clinics in Tennessee US | 38 participants. | Female: n = 24(63) Male: n = 14 (37) | 49.1±9.9 | Latino | Diabetes (not specified the type) | Not mentioned | Convenience sample Focus groups. Data analysis was not reported clearly |
| Lynch *et al.* (2012) [46] | 'To explore low-income minority patients' concepts of diabetes self-management and assess the extent to which patient beliefs correspond to evidence-based recommendations.' | Community and hospital-based safety-net general medicine and diabetes clinics in Chicago. US | 84 participants 49 Mexican American 35 African American | Mexican American: Female: n = 21(43) Male: n = 28 (57) African American: Female: n = 17(48) Male: n = 18 (52) | Mexican American: 55±10.8 African American: 57.6±10.7 | Mexican American African American | Type 2 diabetes | Not mentioned. | Convenience sample Focus groups Grounded theory |

(*Continued*)

**Table 1.** (Continued)

| Author & Year | Study aim (extracted verbatim from the original study) | Setting | Participants | | | | | | | Method of recruitment, data collection & analysis |
|---|---|---|---|---|---|---|---|---|---|---|
| | | | Sample size | Gender/sex reported by participants | Age(mean ±SD)/ median (IQR) | Ethnic minority group: | Type of diabetes | Antidiabetic medication | | |
| Singh *et al.* (2012) [40] | 'To explore barriers to and support systems for optimizing diabetes mellitus in samples of British South Asian and White outpatients with optimal and suboptimal diabetes control.' | Hillingdon Hospital Outpatient clinic. Uxbridge, UK. | 20 participants. (12 British South Asians and 8 British Whites) | British South Asians: Female: n = 6(50) Male: n = 6 (50) British Whites: Female: n = 4(50) Male: n = 4 (50) | 52.6±13.3 | British South Asian British White | Diabetes (Type1 & 2) | Not mentioned. | | Sampling was not reported clearly Semi-structured interviews. Interpretative Phenomenological Analysis |
| Barko *et al.* (2011) [47] | 'To compare and contrast Slavic immigrant women and non-Hispanic, nonimmigrant White American women regarding (a) diabetes and cardiovascular health symptoms; (b) self-management of diabetes, including dietary and physical activity approaches; and (c) perceived educational needs.' | US | 20 participants. Russian-speaking Slavic immigrant American women (10) Non-Hispanic, non-immigrant White American women (10) | Female: n = 20(100) | 63.8 ±10.58 | Russian-speaking Slavic immigrant American women non-Hispanic, non-immigrant White American women | Type 2 diabetes | Not mentioned. | | Convenience sample. Semi-structured interview. Data analysis was not reported clearly |
| Noakes (2010) [41] | 'To explore perceptions of insulin treatment among black African and African-Caribbean people with type 2 diabetes, to gain insight into the barriers to treatment and strategies to overcome resistance.' | Diabetes outpatient clinics at a South London NHS hospital trust UK | 13 participants. | Black African Female: n = 5(38) Male: n = 0 (0) African-Caribbean Female: n = 3(24) Male: n = 5 (38) | 62±8.2 | Black African African-Caribbean | Type 2 diabetes | Insulin and/or oral antidiabetic drugs | | Purposive sampling. Focus groups Data analysis was not reported clearly |
| Ho & James (2006) [53] | 'To determine some of the cultural barriers to initiating insulin therapy among Chinese individuals with type 2 diabetes living in Canada.' | Ambulatory centre for diabetes inToronto, Ontario, Canada. | 5 participants. | Female: n = 1(20) Male: n = 4 (80) | 54.2±9.4 | Chinese | Type 2 diabetes | Insulin therapy | | Sampling was not reported clearly Intensive semi-structured interviews Framework analysis |
| Lawton *et al.* (2005) [42] | 'To explore British Pakistani and British Indian patients' perceptions and experiences of taking oral hypoglycaemic agents (OHAs).' | Primary care and community sources in Edinburgh, Scotland. UK | 32 participants. | Female: n = 17(53) Male: n = 15 (47) | Not reported. | British Pakistani and British Indian | Type 2 diabetes | Oral hypoglycaemic agents. | | Face to face recruitment and snowballing In-depth interviews Grounded theory. |

details of the quality appraisal using the CASP tool for qualitative studies are described in (S3 Table in S1 File).

## Synthesis finding

Three third order constructs (overarching themes developed in this review), with six sub-themes were developed; these comprised of 1) cultural underpinnings, 2) communication, and 3) managing diabetes during visiting home countries. The three overarching themes and sub-themes are outlined in (Fig 3).

The qualitative data synthesis is described in (S4-S6 Tables in S1 File), with each table representing one of the four third order constructs (overarching themes developed in this review). These tables present the direct quotations (first-order constructs) from primary study participants, the original authors' interpretations of the primary findings from the included studies (second-order constructs), and the interpretation from authors of this meta-ethnographic systematic review (third-order constructs) with subthemes.

**1) Cultural underpinnings.** *Perspectives of prescribed medicine and preferences for alternatives*. The preference for alternative medicines was reported mainly as a barrier for adherence to antidiabetic medication amongst ethnic minority communities. For many people, alternative medicine preferences arose from distrust of Western medicine, fear of adverse effects, or dependency [47, 50, 53, 55]. Some participants/people also reported that religious and cultural beliefs in the effectiveness of alternative remedies support their use alongside or instead of antidiabetic medication [38, 46, 47]. Social networks of people from ethnic minority communities also played a role in the use of herbal medicines over prescribed antidiabetic medication, with some people recommending products they have previously tried or have found successful [19, 39, 49, 51, 54].

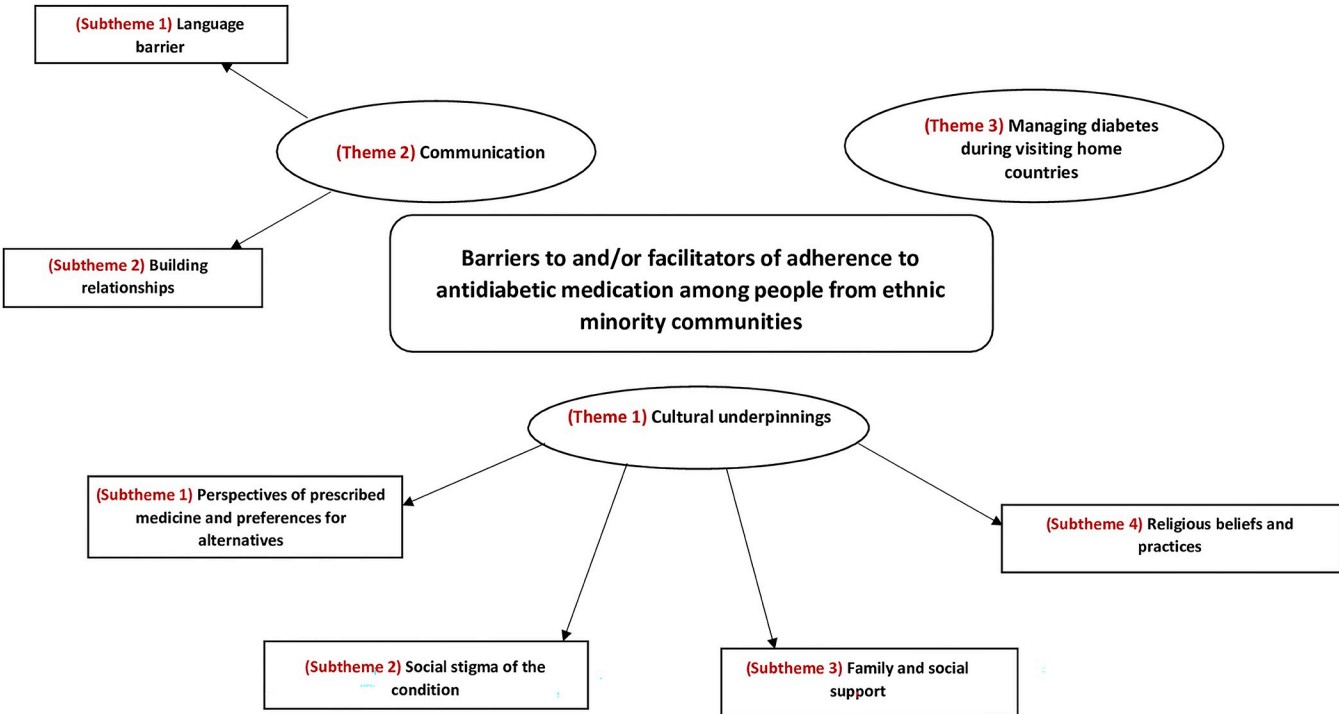

**Fig 3. Developed themes and subthemes for the barriers to and facilitators of adherence to antidiabetic medication among people from ethnic minority communities.**

*'How long has insulin been around?. . . . versus 2000 years of Chinese history. . .'. [53]*

*'Want to be away from all these strong drugs [because of side effects] and make addicted. . . . going to start natural method such as Ayurvedic medication'. [50]*

*'I use Aloe Vera to lower my blood sugar. . . . . .'. [38]*

*'Lots of people in the family have it so they used to tell me. . .. try karela juice. . .'. [39]*

*'. . ..people say, cinnamon is very good for diabetes and tamarind is very good, so one time I start taking those.. '. [49]*

*'Herbal medicine is very effective in treating diabetes, when you take the western medicine in excess you can get other complications '. [55]*

On the other hand, a person's perspectives about prescribed medicines were demonstrated to be both a barrier to, and a facilitator of, adherence among some people from ethnic minority groups. In one study by Lawton *et al.*, one participant discussed views that the antidiabetic medications in their home country were less effective or of lower quality than the medications in their country of residence; in turn, this was seen to create a barrier to adherence to their medications as they returned home for holiday celebrations [42]. In contrast, the same participant described perspectives of trust when considering the effectiveness of antidiabetic medication in their country of residence, which motivated their adherence [42].

*'See, in Pakistan, the medications are not right, they're just a waste of time, waste of money. I mean these [referring to OHAs] are the real stuff. These are what really work'. [42]*

*Social stigma of the condition.* Social stigma was reported to have a negative influence on adherence to antidiabetic medication, namely by participants from Chinese, South Asian and Black minority ethnic communities [19, 40, 41, 53]. Some individuals reported hesitance to disclose their diabetes to their families and community. This affected the use of antidiabetic medication in public, particularly an avoidance of injecting insulin therapies, due to concerns about being stigmatised for drug use [19, 40, 50].

*'Honestly, I intentionally sometimes not take [injections] it in front of people [. . .] I remember I was once taking insulin, and someone got up [. . .] and from far he screamed [. . .] hey heroin!'. [19]*

Moreover, participants in the study by Singh *et al.* reported instructing their relatives with diabetes not to tell anyone about their condition or whether they are taking insulin. Diabetes was considered a sign of physical inadequacy in a social system of South Asian communities affecting marriage prospects in which family elders arranged the majority of marriages [40]. Consequently, the impact of this stigma meant it was difficult for patients to adhere to their medication, particularly during social gatherings.

*'My family views people taking insulin as more of a handicap. . ..'. [53]*

*Family and social support.* Family and social support appeared a crucial facilitator of adherence to antidiabetic medication; in particular, this was noted amongst the studies involving people from African American, South Asian, and Turkish minority communities [18, 20, 40, 54]. Supportive mechanisms were identified as positive influencers too, particularly forms of social support provided by the community centre, religious groups, and other social support

groups [20, 39]. Specifically, peer-led education was deemed beneficial [18, 20, 39, 41]. In South Asian and Turkish communities, the presence of family members also appeared to facilitate a person's adherence to their antidiabetic medication [40, 54]. Some participants indicated that their close family members supported their diabetes self-care by helping them take their medication; this was deemed a facilitator of adherence to antidiabetic medication.

> '.. (. . .) When I travel to Turkey my daughter is always with me. She takes care of my insulin and my pills'. [54]

> '. . .. when I go to the mosque to pray, other people that have diabetes, they talk about it. . .'. [39]

> '. . ..Daughters remind me did you take it?. . . .. The other day, I ate out with friends and then forgot to take my medicine'. [49]

*Religious beliefs and practices.* Religious belief and practices were perceived as both barriers to, and facilitators of, antidiabetic medication adherence in people with South Asian, Black, and Turkish ethnicities [19, 20, 38, 40, 41, 43, 54]. For some participants, they found praying to God was beneficial in finding emotional support for managing their diabetes [20, 38, 40, 41, 43, 54]. However, some people described depending on prayer, instead of medication, for the complete management of diabetes [38, 41, 55]. According to some participants, religious practices such as fasting had a negative impact on adherence to antidiabetic medication since fasting and eating 'light' food (diet high in vegetables and low in carbohydrates and fat)were believed to eliminate the need to take the medications [54].

> 'Prayer is my main key to my strength and managing my diabetes. . .'. [38]

> 'You put everything in prayers. Even though I don't use medication to control my glucose level, I still put other remedies I use in prayer'. [38]

> 'I know prayer can cure diabetes; it all depends on the faith you have'. [55]

> 'There is one-month fasting but not complete; we can use vegetable, not non-vegetable. Light food and I do not use insulin. Only vegetable, so I do not think this will increase my sugar'. [19]

**2) Communication.** *Language barrier.* Language barriers between patients and healthcare providers were seen to negatively influence how people from ethnic minority communities communicate effectively and understand the information received on diabetes management; in turn, this affected their adherence to antidiabetic medication. One of the language-related issues reported across people from ethnic minority communities was the difficulty they encountered in understanding their health providers' consultations about diabetes management [18, 19]. This was specifically reported among people from South Asian ethnic minority community participants who found difficulty in understanding the information provided during consultations with their English-speaking healthcare providers because of their language competency [18].

> 'I sometimes can't understand what the GP/nurse is explaining about my diabetes as they do not speak Urdu'. [18]

Another language-related issue was the difficulty experienced by people from ethnic minority communities in asking their healthcare professionals for information about diabetes

management. A lack of fluency or inability to speak the primary language of their country made it difficult for people from South Asian, Korean American, Latino, and Turkish ethnic minorities to ask or explain their concerns about their diabetes and its management [18, 40, 44, 45, 54]. Moreover, participants from studies by Singh *et al*. and Timsina *et al*. reported that the presence of healthcare providers from their countries of origin who understand their language is preferable when communicating about diabetes [40, 48]. Additionally, the availability of diabetes-related information in only one language is considered a barrier to accessing diabetes information [18, 46]. People from ethnic minority communities were directly affected by all of these issues, which can result in reduced adherence to prescribed antidiabetic medication.

*'I did (want to ask for more information) but I don't know the language'.* [54]

*'Our doctor helps a lot. We are so happy that he can speak and understand our language (Nepalese language)'.* [48]

*Building relationships.* The relationships between patients and healthcare professionals play an essential role in effective communication, which, in turn, is critical in promoting medication adherence. Healthcare providers' time constraints hinder building relationships with patients, resulting in suboptimal counselling on the treatment and thereby hindering medication adherence. Participants from South Asian ethnic minority communities described healthcare professionals' time constraints as a barrier to receiving detailed information during consultations and support from their healthcare providers regarding diabetes management [18, 19, 39]. As a result, people from the same ethnic minority communities also reported dependence on healthcare providers from their home country to seek information about diabetes management [39]. Consequently, inadequate interaction time between patient-healthcare providers led to limited information sharing regarding treatment benefits and listening to patients' concerns about medications, thus negatively influencing adherence to antidiabetic medication.

*'….they won't tell you anything in depth as appointments are limited to only few minutes'.* [18]

*'…you can't blame them because they are seeing so many patients a day, they haven't got the time to spend 20 minutes or half an hour to talk and tell you things….'.* [39]

**3) Managing diabetes during visiting home countries.** The beliefs of people from South Asian and Turkish minority ethnic communities that diabetes being temporarily cured or their blood sugar levels improving in their countries of origin negatively affected adherence to antidiabetic medication [19, 39, 54]. This can be attributed to the strong belief among some participants about the perceived benefits of returning to a hot climate, which they believed improved their diabetes control or provided a temporary cure through sweating [39, 54], leading them to either alter their antidiabetic medication regimen or stoptaking it entirely [19, 39, 54]. Moreover, some participants viewed their cultural holiday in their home countries as a break from taking medication, so they stopped taking their antidiabetic medication [39]. Changing time zones during their travels was also described as a difficulty in adhering to antidiabetic medication regimens [52]. Additionally, the practical issues related to medication storage in native countries of people from ethnic minority communities were acknowledged as a barrier to taking antidiabetic medication during the holidays [39, 41]. A few participants mentioned the inability to store insulin at a suitable temperature due to a lack of electricity

and a refrigerator in their home countries as a barrier to adherence to antidiabetic medication [39, 41]. People from South Asian, African-Caribbean, and Turkish ethnic minority communities reported these barriers while visiting their home countries to adhere to antidiabetic medication.

> *'In Turkey I don't take it (his OHA) at all. I sometimes measure my sugar and it is always lower than (when I am) over here'. [54]*

> *'When I go there my diabetes is gone. . . .'. [39]*

> *'. . ..no electricity for about 8 hours. . .so in the summer it's very difficult and because I take insulin I have nowhere. . .'. [39]*

### Confidence in the synthesised findings

According to GRADE-CERQual methods [37] applied to these review findings, all the sub-themes under the overarching theme 'Cultural underpinnings' were appraised as high confidence except for the subtheme 'Social stigma of the condition' graded as moderate confidence. The second overarching theme 'Communication' was appraised as moderate confidence, and theme 'Managing diabetes during visiting home countries' as high confidence. The confidence ratings indicate that these themes are likely to be reasonable representation of the barriers to and facilitators of adherence to antidiabetic medication among ethnic minority groups in high-income countries. The summary of findings is presented in (S8 Table in S1 File).

## Discussion

This is the first meta-ethnography systematic review to explore the barriers to and facilitators of adherence to antidiabetic medications in people from ethnic minority communities, living in high-income countries. Following a comprehensive search that identified 13,958 potential studies, 20 articles were included in this review and synthesised using a meta-ethnographic approach [34]. The main themes developed around the barriers to and facilitators of adherence to antidiabetic medication among people from minority groups were the following: 1) cultural underpinnings, 2) communication and construction of relationships, and 3) managing diabetes during visiting home countries. Two of the synthesised overarching themes were identified to be both barriers and facilitators in the following subthemes: perspectives of prescribed medicine and preferences for alternatives, and religious beliefs and practices.

The cultural underpinnings of the preference for traditional medicine among people from ethnic minority communities were considered mainly to be a barrier of adherence to antidiabetic medications. This review indicated that negative beliefs and fears about the adverse effects of antidiabetic medication result in a preference for using alternative treatments. These findings are consistent with the broader literature considering the cultural misconceptions of Western medicine in people from South Asian and Black ethnic groups [25, 56, 57]. However, the perception of people from South Asian ethnic minority groups about the higher quality and effectiveness of prescribed medicines in the UK compared to those available in their home countries was demonstrated in this review as contributing to adherence toward antidiabetic medications. Therefore, developing educational programmes tailored to ethnic minority communities might be worthwhile to correct the negative beliefs about their antidiabetic medication and alternative medicine.

The social stigma related to diabetes in people from ethnic minority communities was also considered to be a significant barrier to adherence to antidiabetic medication. Stigma affected

adherence in these groups if their diabetes was disclosed to their communities or even to their families, resulting in skipping a dose or in difficulty taking medications during social gatherings. This theme was supported by the findings of a qualitative review that explored beliefs about medicines in people of South Asian origin with diabetes and cardiovascular diseases, which concluded that the cultural stigma associated with diabetes presented a barrier of adherence to medication [58]. Social stigma has been linked to misconceptions, lack of awareness, and local health beliefs about diabetes in some ethnic groups [41, 59, 60]. Increasing the awareness and management of diabetes among people from minority ethnic communities could reduce the fear of stigma and improve medication adherence.

This review identified that social and family supports played a crucial role in enhancing adherence to antidiabetic medication among people from minority ethnic groups. Social networks of South Asian and Black ethnic groups support increasing self-management knowledge of diabetes through social groups and faith-based organisations. Moreover, a family member in people from South Asian ethnic minority communities played a significant role in enabling medication adherence. The positive association between medication adherence and social support was also reported in a meta-analysis examining the impact of patients' social supports on adherence to medications in hypertensive patients [61]. Faith-based organisations are helpful as a support system for the dissemination of information related to diabetes and its management targeting ethnic minority groups.

One interesting finding of this review is that religious beliefs played a vital role in helping people from ethnic minority communities to adhere to their antidiabetic medication. These beliefs contributed to emotional support for taking medication in people from South Asian minority communities and helped people from Black minority communities in their decisions to take antidiabetic medications. This finding is contrary to previous reviews, which suggested that patients' religious beliefs did not help with type 2 diabetes management among people from the Caribbean region [62].This inconsistency between the two reviews may be traced to different religious groups and beliefs. However, religion-related practices such as fasting were identified in this review as a barrier to adherence to antidiabetic medication, mainly among people from South Asian and Turkish minority groups. Members of these groups tend to modify their doses of or stop taking antidiabetic medication altogether due to their belief that fasting and eating light food eliminates the need for medication.

Language barriers were identified by this review to impede effective communication and relationships between healthcare providers and people with diabetes from ethnic minority groups. Lack of language concordance results in difficulty understanding the provided information and having questions about diabetes management answered as well as reluctance to seek more information. All of these factors affect medication adherence in these groups. This finding is supported by a recent systematic review, which found that language barriers led to miscommunication between healthcare professional and patients [63]. Another study, conducted in the United States, examined the influence of language concordance between patients and physicians on medication adherence, and revealed that language concordance was significantly associated with medication adherence [64]. Limited consultation time with medical professionals also hindered adherence to antidiabetic medication among people from minority ethnic groups. This finding was also reported by Jin *et al.*, who found that patient motivation for adherence to therapy can be compromised by the limited time spent with healthcare professionals [65].

People from minority ethnic communities indicated in this review that they have difficulty adhering to their antidiabetic medication regimens in their home countries. This is mainly attributed to casual beliefs that the hot climate and low stress they find at home result in their diabetes being controlled well without taking medication [39, 54]. Additionally, the lack of

availability of facilities to store insulin in the native countries of people from South Asian and African Caribbean minorities was reported as a difficulty in adherence to insulin [39, 41]. Given the negative beliefs about diabetes management among people from minority ethnic communities during their holidays, a pretravel tailored education programme could improve patients' adherence to antidiabetic medications and, accordingly, their health outcomes.

## Strengths

This review has a number of strengths. Firstly, it is the first meta-ethnography systematic review to provide a comprehensive qualitative synthesis of the barriers to and facilitators of adherence to antidiabetic medications in people from ethnic minority communities in high-income countries. Another strength of this review is that the findings have been reported according to the PRISMA guidelines. Furthermore, the systematic search was not limited by language or date, and complementary searches, such as backward and forward citation searching and grey literature searches, were undertaken to ensure a robust literature search (S1 Text).

## Limitations

Despite the strengths, some limitations should be acknowledged. One limitation is that few qualitative studies explored the perspective of healthcare professionals on the barriers to and facilitators of adherence to antidiabetic medications. Therefore, future studies are needed to explore that from healthcare professionals' perspectives. Another limitation is that since this meta-ethnography focused primarily on minority ethnic groups living in high-income countries, and more than half of the included studies were conducted in the UK and the United States, our findings may not be generalisable to minority ethnic communities living in other high-income countries. The review's findings may not also apply to different types of diabetes and second/third-generation migrants, as the majority of included studies focused on people with type 2 diabetes and migrants or refugees. Another limitation is that the titles and abstracts were screened by only one reviewer, which can introduce subjective bias and increase the risk of missing relevant studies. The last limitation for this review that all ethnic minority groups have considered as one group, overlooking the potential variations and complexities among different groups. In future studies, it would be valuable to investigate the barriers to and facilitators of adherence to antidiabetic medications in each ethnic group to gain a deeper understanding of the diverse experiences within each ethnic minority group.

## Implications of findings for practice, public health and future research

The review findings on the barriers and facilitators of adherence to antidiabetic medication among people from minority ethnic communities in high-income countries have significant implications for clinical practice and public health. Healthcare professionals should recognize the impact of cultural-related factors on medication adherence among minority ethnic communities. By providing culturally sensitive care, healthcare professionals can build trust and rapport with patients, resulting in improved patient-provider communication and greater adherence to antidiabetic medication. Understanding the barriers and facilitators of medication adherence can also lead to developing patient-centred approaches, tailoring medication adherence intervention based on individual needs and cultural backgrounds. It is also important for healthcare professionals to receive communication training, especially in cross-cultural situations, to improve patient-provider interactions and medication adherence. In public health, culturally and socially relevant interventions tailored to the needs of minority ethnic communities are crucial in improving adherence rates and diabetes outcomes. Addressing medication adherence disparities contributes to health equity promotion, necessitating public

health policies prioritising culturally competent interventions. Further research is also essential to understand the cultural underpinnings influencing medication adherence within different minority ethnic communities. Qualitative research helps to understand the experiences and perspectives of individuals from minority ethnic communities regarding medication adherence. This research may lead to developing targeted interventions tailored to these populations based on their cultural context. Moreover, intervention studies can help in developing these interventions and examine their effectiveness in improving adherence among people from minority ethnic communities.

## Conclusion

This meta-ethnographic systematic review is one of the first to explore barriers to and facilitators of adherence to antidiabetic medication in people from ethnic minority communities in high-income countries. A number of barriers and facilitators have been identified among people from ethnic minority communities. The findings of this meta-ethnographic synthesis provide an explanation of why people from ethnic minority communities reported a lower adherence rate to antidiabetic medications compared to the majority population. Therefore, tailored medication adherence interventions to overcome these barriers are vital for improving the diabetes care of people from these groups.

## Supporting information

**S1 File.**
(ZIP)

**S1 Text. Database search terms.**
(DOCX)

## Author Contributions

**Conceptualization:** Rayah Asiri, Adam Todd, Andy Husband.

**Data curation:** Rayah Asiri, Anum Iqbal, Adam Todd, Andy Husband.

**Formal analysis:** Rayah Asiri, Anna Robinson-Barella, Adam Todd, Andy Husband.

**Investigation:** Rayah Asiri, Anna Robinson-Barella, Adam Todd, Andy Husband.

**Methodology:** Rayah Asiri, Anna Robinson-Barella, Adam Todd, Andy Husband.

**Project administration:** Adam Todd, Andy Husband.

**Supervision:** Anna Robinson-Barella, Adam Todd, Andy Husband.

**Writing – original draft:** Rayah Asiri.

**Writing – review & editing:** Anna Robinson-Barella, Adam Todd, Andy Husband.

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
