## [Decision Letter · Decision Letter 0]

26 Jun 2023

PONE-D-23-15435Understanding the influence of ethnicity on adherence to antidiabetic medication: Meta-ethnography and systematic reviewPLOS ONE

Dear Dr. Asiri,

Thank you for submitting your manuscript to PLOS ONE. After careful consideration, we feel that it has merit but does not fully meet PLOS ONE’s publication criteria as it currently stands. Therefore, we invite you to submit a revised version of the manuscript that addresses the points raised during the review process.

We look forward to receiving your revised manuscript.

Kind regards,

Anil Gumber, Ph.D.

Academic Editor

PLOS ONE

Journal Requirements:

   "All data related to this study are included in this published article (and its supplementary 

information files."

4. We note that this manuscript is a systematic review or meta-analysis; our author guidelines therefore require that you use PRISMA guidance to help improve reporting quality of this type of study. Please upload copies of the completed PRISMA checklist as Supporting Information with a file name “PRISMA checklist”.

Additional Editor Comments:

See attached comments from reviewers.

Reviewers' comments:

Reviewer's Responses to Questions

**Comments to the Author**

1. Is the manuscript technically sound, and do the data support the conclusions?

Reviewer #1: Yes

Reviewer #2: Yes

2. Has the statistical analysis been performed appropriately and rigorously? 

Reviewer #1: N/A

Reviewer #2: N/A

3. Have the authors made all data underlying the findings in their manuscript fully available?

Reviewer #1: No

Reviewer #2: Yes

4. Is the manuscript presented in an intelligible fashion and written in standard English?

Reviewer #1: Yes

Reviewer #2: Yes

5. Review Comments to the Author

Reviewer #1: Introduction

1. ‘’Across a number of high-income countries, lower rates of adherence to antidiabetic medications has been reported amongst minority ethnic groups. [12-16]’’; Such as…….

Methods

Database:

1. I think the list of databases is not complete. Other relevant database such as Global Health was not searched – that could potentially yield relevant articles for this review. Is there any specific reason why this database was not searched?

Study selection and screening:

1. What platform (s) did the authors use for screening of articles? This should be stated.

2. For rigour and transparency two independent reviewers are expected to screen titles and abstracts. Why did only one reviewer screen titles and abstracts?

3. ‘’The screenings of the articles’ full text were undertaken by RA and checked in full by authors AKH and AR-B.’’ This statement is not clear. Did the named authors independently screen the full text articles? If so, this should be explicitly stated.

Study reading, data extraction and quality appraisal

1. Two reviewers should independently assess the quality of systematic review using the Assessing the Methodological Quality of Systematic Reviews (AMSTAR) tool – this should be submitted with the manuscript.

2. Did authors work independently to extract data, or was any process in place for obtaining or confirming data from study investigators?

3. Confidence in the evidence: The authors should use the Confidence in the Evidence from Reviews of Qualitative research (CERQual) tool to assess the confidence of the key findings of this review. The authors could report or summarise findings in a table as part of review.

Discussion

1. Strengths and limitations of the review should have a separate subheading.

2. The authors should provide a separate subheading under the discussion section to discuss in detail the implications of their findings for practice, public health and future research.

Reviewer #2: Thank you for the opportunity to review this manuscript, which describes a systematic review of evidence on the influence of ethnicity on adherence to antidiabetic medication. Although rather descriptive, the meta-ethnography has been conducted competently and a clear rationale is giving for using this approach. I have very few suggested areas for improvement. The first relates to the methods section, which mentions grey literature (line 100) but provides no detail on how this was reviewed. Which grey literature sources were searched, what terms/strategy was used, etc? The discussion section mentions ‘backward and forward citation chaining’ (lines 460-461), which needs to be described in the methods. It would also be helpful to provide a brief rationale for excluding mixed methods studies and qualitative survey data from the review.

The second relates to the discussion, which makes no mention of the fact that most included studies involved people with type 2 diabetes and many also focused on migrants or refugees. The authors may wish to comment on this in the results and/or discussion sections. At the very least, they should acknowledge that the findings may not be generalisable to those with other types of diabetes or to second and third generation migrants.

Other suggested areas for improvement/minor revisions:

Abstract:

Lines 29-30: The study context needs specifying from the outset (UK? High income countries?)

Lines 43-45: The results section needs more detail, rather than listing the three themes

Introduction

Lines 53-53: It would be helpful to mention the different types of diabetes here or later in the paragraph

Lines 85-88: This sentence is rather convoluted and needs re-wording

Methods

Lines 134-135: Need to specify that the CASP tool for qualitative studies was used

Lines 137-138: Convoluted, needs re-wording

Results

Study characteristics: There are some errors in this paragraph, e.g. the list of countries does not include Nepal, while the list of ethnicities does not include Black African, Latino, Slavic (all mentioned in table 1).

A diagram to illustrate the various themes and sub-themes would be useful here.

Lines 344-345: Needs re-wording: ‘which, in turn, negatively influence adherence’

Line 356: Alter, not altered

Line 357: Stop, not stopped

Line 361: ‘some ethnic minorities’ native countries’ – it would be better to use person first language, e.g. people from ethnic minority communities

371-372: The final quote is missing a reference

Discussion

Line 421: ‘people from ethnic minority communities’

Line 426: ‘may be traced’

Conclusion

Line 479: ‘ethnic minority communities’

6. PLOS authors have the option to publish the peer review history of their article (what does this mean?). If published, this will include your full peer review and any attached files.

Reviewer #1: No

Reviewer #2: **Yes: **Shelina Visram

<quillbot-extension-portal></quillbot-extension-portal>

---

## [Author Response · Author response to Decision Letter 0]

21 Sep 2023

Academic editor comments Authors’ Response

1. Please ensure that your manuscript meets PLOS ONE's style requirements, including those for file naming. We have verified that all formatting and style requirements have been met.

 "All data related to this study are included in this published article (and its supplementary information files."

Thanks for the comment. We have changed the statements.

Our updated data availability statement is as follows:

 "All data related to this study are included in this published article (and its supplementary information files.) This does not alter our adherence to PLOS ONE policies on sharing data and materials "

Our updated competing interests’ statement is as follows:

"All authors declare that they have no conflicts of interest. "

Thanks for your kind reminders, however, all data related to this study are included in this published article and its supporting information files. 

Our updated data availability statement is as follows:

"All data related to this study are included in this published article (and its supplementary information files. This does not alter our adherence to PLOS ONE policies on sharing data and materials "

5. We note that this manuscript is a systematic review or meta-analysis; our author guidelines therefore require that you use PRISMA guidance to help improve reporting quality of this type of study. Please upload copies of the completed PRISMA checklist as Supporting Information with a file name “PRISMA checklist”. 

Thank you for your comment. The PRISMA checklist has already been included as Supporting Information in the supplementary files.

[Supporting information file: S1 Table and S2 Table]

6- Have the authors made all data underlying the findings in their manuscript fully available?

The PLOS Data policy requires authors to make all data underlying the findings described in their manuscript fully available without restriction, with rare exception (please refer to the Data Availability Statement in the manuscript PDF file). The data should be provided as part of the manuscript or its supporting information or deposited to a public repository. For example, in addition to summary statistics, the data points behind means, medians and variance measures should be available. If there are restrictions on publicly sharing data—e.g. participant privacy or use of data from a third party—those must be specified.

Reviewer #1: No 

Thank you for your comment, however, all data related to this study are included in this published article and its supporting information files.

Response to Reviewer 1

Introduction 

1- ‘’Across a number of high-income countries, lower rates of adherence to antidiabetic medications has been reported amongst minority ethnic groups. [12-16]’’; Such as……. 

Thank you very much for the comment. We have made revisions accordingly as follow:

[Page:4, Line:84 - 88]:

"In a number of high-income countries such as the United States, the United Kingdom, New Zealand, Singapore, and Canada, lower rates of adherence to antidiabetic medications have been reported amongst minority ethnic groups"

Methods

Database:

1. I think the list of databases is not complete. Other relevant database such as Global Health was not searched – that could potentially yield relevant articles for this review. Is there any specific reason why this database was not searched? 

Thank you for your comment. We appreciate your suggestion to search additional databases such as the Global Health database. However, we would like to provide a justification for our decision to search specific databases, including Medline, Embase, CINAHL, and PsycINFO. Our choice of these databases was based on careful consideration of their scope, relevance to the research topic, and the availability of relevant literature. We sincerely appreciate your valuable suggestion, which has prompted us to reconsider the Global Health database. Although we did not include the Global Health database in our initial search, we recognize the potential benefit of expanding our search to this database. In light of your suggestion, we have conducted a supplementary search in the Global Health database and identified one additional article that meet our inclusion criteria [ S1 Text]. We carefully assessed this newly identified articles for relevance to our systematic review and included it in our analysis: 

[Page:2, Line: 37] , [Page:3, Line: 47 ] , [Page:6, Line: 117 - 122 ] ,[Page:10, Line: 218 - 223] , [Page:10, Line: 227-230] ,[Page:14, Table1], [Page:21, Line: 245 - 247 ], [Page:22, Line: 284 -285 ], [Page:25, Line: 345 ], [Figure 2], [ S3 Table ], [Page:3, S4 Table ], and [Page:7, S4 Table ]..

Study selection and screening:

1. What platform (s) did the authors use for screening of articles? This should be stated. 

Thank you very much for the comment. We have made revisions accordingly 

as follows:

[Page:7, Line:148 -149]:

"All citations were exported to EndNote 20 reference manager software to manage duplicate studies, and the screening process.[1] "

2. For rigour and transparency two independent reviewers are expected to screen titles and abstracts. Why did only one reviewer screen titles and abstracts? 

Thank you for your comment. We do accept that the gold standard approach would be for two reviewers to independently screen titles and abstracts. On this occasion, for reasons of practicality we employed single reviewer checking at the title and abstract stage. To ensure a rigorous approach, if there were any doubts about the paper at the title and abstract stage it was put through to the full screening stage. At the stage, the decision to include or exclude the paper was independently checked by another reviewer. 

Moreover, while we recognize the advantages of involving multiple reviewers, we also note that other reviews published in PLOS ONE have established a precedent by employing a similar approach.[2-12] We added (screening the title and abstract by one reviewer) as a limitation for this review as follows: 

[Page:32, Line:533 -535]:

"Another limitation is that the titles and abstracts were screened by only one reviewer, which can introduce subjective bias and increase the risk of missing relevant studies. "

3. ‘’The screenings of the articles’ full text were undertaken by RA and checked in full by authors AKH and AR-B.’’ This statement is not clear. Did the named authors independently screen the full text articles? If so, this should be explicitly stated. 

Thank you very much for the reminder. We have made revisions accordingly 

as follows:

[Page:7, Line:152]:

"The screenings of the articles’ full text were undertaken by RA and checked independently in full by authors AKH or AR-B. "

Study reading, data extraction and quality appraisal

1. Two reviewers should independently assess the quality of systematic review using the Assessing the Methodological Quality of Systematic Reviews (AMSTAR) tool – this should be submitted with the manuscript. 

Thank you for your comment. We appreciate your suggestion to utilize the AMSTAR tool to assess systematic review quality. However, while the AMSTAR tool is widely used for evaluating the methodological quality of systematic reviews, it is primarily designed for quantitative systematic reviews that include randomized controlled trials and other quantitative study designs.[13, 14] 

On the other hand, our systematic review focuses on qualitative evidence synthesis, which involves synthesizing qualitative primary research studies rather than quantitative data. For example, question 4 from the AMSTAR checklist states: “Did the review authors use a comprehensive literature search strategy?”. While this is an important consideration when assessing systematic reviews, this type of question is not relevant for our review given we are focusing on primary qualitative literature. As such, the AMSTAR tool may not fully capture the specific methodological considerations and criteria relevant to our primary qualitative evidence synthesis.

2. Did authors work independently to extract data, or was any process in place for obtaining or confirming data from study investigators? 

Thanks for your question. Data extraction for the information about the author's details, study aims, setting, and participant demographics was done by a single reviewer. Two reviewers did independently extract the data related to study findings (including original participant quotes and author interpretations).

3. Confidence in the evidence: The authors should use the Confidence in the Evidence from Reviews of Qualitative research (CERQual) tool to assess the confidence of the key findings of this review. The authors could report or summarise findings in a table as part of review. 

Thank you very much for the comment. We sincerely appreciate your valuable suggestion regarding the incorporation of the Confidence in the Evidence from Reviews of Qualitative research tool to assess the confidence of the key findings in our review. we agree that utilizing the CERQual tool would significantly enhance the transparency and credibility of our qualitative research review. In response to your suggestion, we have summarized the confidence levels associated with each key finding in the method and result section as follows: 

Abstract: 

[Page:2, Line:42-45]:

"The Grading of Recommendations Assessment, Development and Evaluation Confidence in the Evidence from Reviews of Qualitative research (GRADE-CERQual) approach was used to assess the Confidence in the review findings. "

[Page:3, Line:52-53]:

"Based on the GRADE-CERQual assessment, we had mainly moderate- and high-confidence findings. "

Method: 

[Page:9-10, Line:207-214]:

"Confidence in the synthesised findings

The confidence in this review's findings was assessed by the GRADE-CERQual approach (Confidence in the Evidence from Reviews of Qualitative Research). A GRADE-CERQual assessment considers four main components: the methodological limitations of included studies, the coherence of the qualitative evidence synthesis findings, the adequacy of data used to support the review finding, and the relevance of the included studies to the review question. Based on the results of CERQual, we classified overall confidence into three levels: high, moderate, and low, and presented the results in a summary of findings table. "

Result: [Page:28, Line:422 -431], [S8 Table]

"Confidence in the synthesised findings

According to GRADE-CERQual methods [15] applied to these review findings, all the subthemes under the overarching theme ‘Cultural underpinnings’ were appraised as high confidence except for the subtheme ‘Social stigma of the condition’ graded as moderate confidence. The second overarching theme ‘Communication’ was appraised as moderate confidence, and theme ‘Managing diabetes during visiting home countries’ as high confidence. The confidence ratings indicate that these themes are likely to be reasonable representation of the barriers to and facilitators of adherence to antidiabetic medication among ethnic minority groups in high-income countries. The summary of findings is presented in (S8 Table). "

Discussion

1. Strengths and limitations of the review should have a separate subheading. 

Thank you for this suggestion. We added a separate subheading for the strengths and limitations in the discussion section: [Page:31, Line:514], [Page:32, Line:523].

2. The authors should provide a separate subheading under the discussion section to discuss in detail the implications of their findings for practice, public health and future research. Thanks for the suggestion. We added a separate subheading under the discussion section to discuss the implications of the review findings for practice, public health and future research, as follows:

[Page:33, Line:541 -564].

"Implications of findings for practice, public health and future research

The review findings on the barriers and facilitators of adherence to antidiabetic medication among people from minority ethnic communities in high-income countries have significant implications for clinical practice and public health. Healthcare professionals should recognize the impact of cultural-related factors on medication adherence among minority ethnic communities. By providing culturally sensitive care, healthcare professionals can build trust and rapport with patients, resulting in improved patient-provider communication and greater adherence to antidiabetic medication. Understanding the barriers and facilitators of medication adherence can also lead to developing patient-centred approaches, tailoring medication adherence intervention based on individual needs and cultural backgrounds. It is also important for healthcare professionals to receive communication training, especially in cross-cultural situations, to improve patient-provider interactions and medication adherence. In public health, culturally and socially relevant interventions tailored to the needs of minority ethnic communities are crucial in improving adherence rates and diabetes outcomes. Addressing medication adherence disparities contributes to health equity promotion, necessitating public health policies prioritising culturally competent interventions. Further research is also essential to understand the cultural underpinnings influencing medication adherence within different minority ethnic communities. Qualitative research helps to understand the experiences and perspectives of individuals from minority ethnic communities regarding medication adherence. This research may lead to developing targeted interventions tailored to these populations based on their cultural context. Moreover, intervention studies can help in developing these interventions and examine their effectiveness in improving adherence among people from minority ethnic communities. "

Response to Reviewer 2 

1- Thank you for the opportunity to review this manuscript, which describes a systematic review of evidence on the influence of ethnicity on adherence to antidiabetic medication. Although rather descriptive, the meta-ethnography has been conducted competently and a clear rationale is giving for using this approach.

Thank you very much for the comment.

2- The first relates to the methods section, which mentions grey literature (line 100) but provides no detail on how this was reviewed. Which grey literature sources were searched, what terms/strategy was used, etc? The discussion section mentions ‘backward and forward citation chaining’ (lines 460-461), which needs to be described in the methods. 

Thank you for your question and suggestion. We searched the grey literature using the following platforms: (OpenGrey, Google search the top 150 hits, and EThOS), and theses search term were used :( diabetes and qualitative study and medication adherence and/or ethnicity). In light of your question regarding grey literature, we have added the searched grey literature to the result as follows:

[Page:6, Line:119 -120]:

"To identify all relevant publications, grey literature (via searching OpenGrey, the top 150 Google search hits , and EThOS) .."

[S1 Text]:

"Grey literature search terms

(Diabetes and Qualitative study and Medication adherence and/or Ethnicity) "

Regarding (backward and forward citation), We have made revisions accordingly as follows:

[Page:6, Line:120 -122]:

"hand searches of the reference lists of all included studies and relevant systematic reviews, and the citations of all included studies, using the citations provided by Google scholar were performed. "

3-It would also be helpful to provide a brief rationale for excluding mixed methods studies and qualitative survey data from the review. 

Thank you for your valuable suggestions. We excluded mixed methods studies and qualitative survey data from this review as we used a metaethnography approach which is unlike other qualitative synthesis approaches such as thematic synthesis and interpretative synthesis, it focuses exclusively on the inclusion of primary qualitative studies. The fundamental principles and epistemological underpinnings of meta-ethnography emphasize the synthesis of qualitative data from multiple primary studies to generate higher-order interpretations and theories.[16]

As per your suggestion, we have added the rationale for excluding mixed methods studies and qualitative survey data from this review to the method section as follows:

[Page:7, Line:144 -145]:

"study types that were mixed method and quantitative studies as the meta-ethnographic approach's exclusively focus on inclusion of qualitative studies[16] "

4-The second relates to the discussion, which makes no mention of the fact that most included studies involved people with type 2 diabetes and many also focused on migrants or refugees. The authors may wish to comment on this in the results and/or discussion sections. At the very least, they should acknowledge that the findings may not be generalisable to those with other types of diabetes or to second and third generation migrants. 

Thank you for your comment. We added it to the discussion section as a limitation for this review as follows: 

[Page:32, Line:528 -530]:

"The review's findings may not also apply to different types of diabetes and second/third-generation migrants, as the majority of included studies focused on people with type 2 diabetes and migrants or refugees.

 "

5-Other suggested areas for improvement/minor revisions:Abstract Lines 29-30: The study context needs specifying from the outset (UK? High income countries?) 

Thank you very much for the reminder. We have made revisions accordingly 

as follows:

[Page:2, Line:29 -30]:

"A high prevalence of diabetes and diabetes-related complications in people from minority ethnic communities in high income countries is of significant concern."

Lines 43-45: The results section needs more detail, rather than listing the three themes 

Thanks for the comment. We have made revisions accordingly 

as follows:

[Page:3, Line:47 - 53]:

"Of 13,994 citations screened, 21 studies that included primary qualitative studies were selected, each of which involved people from minority ethnic communities from eight high income countries. This qualitative evidence synthesis has identified three overarching themes around the barriers to and facilitators of adherence to antidiabetic medication among ethnic minority groups.: 1) cultural underpinnings, 2) communication and building relationships, and 3) managing diabetes during visiting home countries. Based on the GRADE-CERQual assessment, we had mainly moderate- and high-confidence findings."

Introduction Lines 53-53: It would be helpful to mention the different types of diabetes here or later in the paragraph 

Thank you for your suggestion. We added the different types of diabetes to the introduction as follows: 

[Page:3, Line:65 - 67]:

"There are several types of diabetes, including type 1, type 2, maturity-onset diabetes of the young, gestational diabetes, neonatal diabetes, and secondary causes due to endocrinopathies and steroids use.[17] "

Lines 85-88: This sentence is rather convoluted and needs re-wording 

Thanks for the comment. We have made revisions accordingly as follows:

[Page:5, Line:101 - 107]:

"This review utilized a meta-ethnographic qualitative synthesis approach to explore the barriers to, and facilitators of, adherence to antidiabetic medications experienced by people from minority ethnic communities in high-income countries. "

MethodsLines 134-135: Need to specify that the CASP tool for qualitative studies was used 

Thank you very much for the reminder. We have made revisions accordingly 

as follows:

[Page:7, Line:161]:

"Quality assessment was conducted by (RA and AI) using the Critical Appraisal Skills Programme (CASP) tool for qualitative research[18]".

Lines 137-138: Convoluted, needs re-wording 

Thanks for the comment. We have made revisions accordingly as follows:

[Page:8, Line:163 -165]:

"This review used the meta-ethnographic approach described originally by Noblit and Hare[34], and commonly used in healthcare research[26, 27, 35]. The seven phases of the meta-ethnography approach[34] are listed in (Fig 1). "

Results

Study characteristics: There are some errors in this paragraph, e.g. the list of countries does not include Nepal 

Thank you for your comment. We greatly value your observation regarding the list of countries mentioned in the Study Characteristics paragraph. We would like to take this opportunity to clarify the nature of the study in question. [19] It is indeed a multisite study, encompassing data collection in both high-income (Australia) and middle-income (Nepal) countries. However, it's important to note that for the purposes of this particular review, our focus centres exclusively on the qualitative data collected from people belonging to ethnic minority communities in high-income countries. As a result, we didn’t include Nepal in the list of countries.

Considering your observation, we added a clarification regarding that to the method section as follows:

[Page:6, Line:134 -136]:

" In the case of studies conducted in “mixed” countries (high-income and low or middle-income countries), the data exclusively from the high-income countries were included. "

The list of ethnicities does not include Black African, Latino, Slavic (all mentioned in table 1). 

Thank you very much for the reminder. We have made revisions accordingly as follows:

[Page:10-11 , Line:232-234]:

"Minority ethnic groups that were reported across the included studies were: Asian, African American, Mexican American, African Caribbean, Turkish, Māori, Black African, Latino, Slavic and Pacific. "

A diagram to illustrate the various themes and sub-themes would be useful here. 

Thank you for your suggestion. We added illustrative diagram (Fig 3) for the themes and sub-themes and made revisions as follows:

[Page:21, Line:254-258]:

"The three overarching themes and subthemes are outlined in (Fig 3). "

Lines 344-345: Needs re-wording: ‘which, in turn, negatively influence adherence’ 

Thanks for the comment. We have made revisions accordingly as follows:

[Page:26, Line:391-392]:

".., thus negatively influencing adherence to antidiabetic medication. "

Line 356: Alter, not altered 

Thanks for the comment. We have made revisions accordingly as follows:

[Page:27, Line:404]:

"...leading them to either alter.."

Line 357: Stop, not stopped 

Thanks for the comment. We have made revisions accordingly as follows:

[Page:27, Line:405]:

"...or stop taking it.. "

Line 361: ‘some ethnic minorities’ native countries’ – it would be better to use person first language, e.g. people from ethnic minority communities 

Thanks for the comment. We have made revisions accordingly as follows:

[Page:27, Line:409-410]:

"Additionally, the practical issues related to medication storage in native countries of people from ethnic minority communities were acknowledged as a barrier to taking antidiabetic medication during the holidays. "

371-372: The final quote is missing a reference 

Thank you very much for the reminder. We added the missing reference.

Discussion Line 421: ‘people from ethnic minority communities’ 

Thanks for the comment. We have made revisions accordingly as follows:

[Page:30, Line:479]:

"..people from ethnic minority communities.. "

Line 426: ‘may be traced’ 

Thanks for the comment. We have made revisions accordingly as follows:

[Page:30, Line:485]:

"..may be traced to.."

Conclusion Line 479: ‘ethnic minority communities’ 

Thanks for the comment. We have made revisions accordingly as follows:

[Page:34, Line:570]:

"..people from ethnic minority communities.. "

References:

1.Gotschall T. EndNote 20 desktop version. Journal of the Medical Library Association: JMLA. 2021;109(3):520.

2.Zaman M, Koski A. Child marriage in Canada: A systematic review. PLoS One. 2020;15(3):e0229676.

3.Broadhurst D, Cooke M, Sriram D, Gray B. Subcutaneous hydration and medications infusions (effectiveness, safety, acceptability): A systematic review of systematic reviews. PLoS One. 2020;15(8):e0237572.

4.Marsh-Feiley G, Eadie L, Wilson P. Telesonography in emergency medicine: a systematic review. PloS one. 2018;13(5):e0194840.

5.Choudhry V, Dayal R, Pillai D, Kalokhe AS, Beier K, Patel V. Child sexual abuse in India: A systematic review. PloS one. 2018;13(10):e0205086.

6.Talevski J, Wong Shee A, Rasmussen B, Kemp G, Beauchamp A. Teach-back: A systematic review of implementation and impacts. PloS one. 2020;15(4):e0231350.

7.McKay AJ, Patel RK, Majeed A. Strategies for tobacco control in India: a systematic review. PLoS One. 2015;10(4):e0122610.

8.O’Connor AM, Cousins G, Durand L, Barry J, Boland F. Retention of patients in opioid substitution treatment: a systematic review. PloS one. 2020;15(5):e0232086.

9.Hernandez-Suarez G, Saha D, Lodroño K, Boonmahittisut P, Taniwijaya S, Saha A, et al. Seroprevalence and incidence of hepatitis A in Southeast Asia: A systematic review. Plos one. 2021;16(12):e0258659.

10.Antoun M, Edwards KM, Sweeting J, Ding D. The acute physiological stress response to driving: A systematic review. PLoS one. 2017;12(10):e0185517.

11.Filoso S, Bezerra MO, Weiss KC, Palmer MA. Impacts of forest restoration on water yield: A systematic review. PloS one. 2017;12(8):e0183210.

12.Asiri R, Todd A, Robinson-Barella A, Husband A. Ethnic disparities in medication adherence? A systematic review examining the association between ethnicity and antidiabetic medication adherence. Plos one. 2023;18(2):e0271650.

13.Shea BJ, Reeves BC, Wells G, Thuku M, Hamel C, Moran J, et al. AMSTAR 2: a critical appraisal tool for systematic reviews that include randomised or non-randomised studies of healthcare interventions, or both. bmj. 2017;358.

14.Shea BJ, Grimshaw JM, Wells GA, Boers M, Andersson N, Hamel C, et al. Development of AMSTAR: a measurement tool to assess the methodological quality of systematic reviews. BMC medical research methodology. 2007;7(1):1-7.

15.Lewin S, Booth A, Glenton C, Munthe-Kaas H, Rashidian A, Wainwright M, et al. Applying GRADE-CERQual to qualitative evidence synthesis findings: introduction to the series. BioMed Central; 2018. p. 1-10.

16.Sattar R, Lawton R, Panagioti M, Johnson J. Meta-ethnography in healthcare research: a guide to using a meta-ethnographic approach for literature synthesis. BMC health services research. 2021;21:1-13.

17.Sapra A, Bhandari P. Diabetes. StatPearls. Treasure Island (FL): StatPearls Publishing Copyright © 2023, StatPearls Publishing LLC.; 2023.

18.CASP C. CASP qualitative checklist. Critical Appraisal Skills Programme. 2018.

19.Sapkota S, Brien J-aE, Aslani P. Nepalese patients’ anti-diabetic medication taking behaviour: an exploratory study. Ethnicity & health. 2018;23(7):718-36.

---

## [Editor Report · Decision Letter 1]

25 Sep 2023

Understanding the influence of ethnicity on adherence to antidiabetic medication: Meta-ethnography and systematic review

PONE-D-23-15435R1

Dear Dr. Husband,

We’re pleased to inform you that your manuscript has been judged scientifically suitable for publication and will be formally accepted for publication once it meets all outstanding technical requirements.

Kind regards,

Anil Gumber, Ph.D.

Academic Editor

PLOS ONE

Additional Editor Comments (optional):

Comment to reviewers are adequately addressed. Thanks
---

## [Editor Report · Acceptance letter]

4 Oct 2023

PONE-D-23-15435R1 

Understanding the influence of ethnicity on adherence to antidiabetic medication: Meta-ethnography and systematic review 

Dear Dr. Husband:

I'm pleased to inform you that your manuscript has been deemed suitable for publication in PLOS ONE. Congratulations! Your manuscript is now with our production department. 

Kind regards, 

on behalf of

Dr. Anil Gumber 

Academic Editor

PLOS ONE